# A Facile Fabrication of Lysosome-Targeting pH Fluorescent Nanosensor Based on PEGylated Polyester Block Copolymer

**DOI:** 10.3390/polym14122420

**Published:** 2022-06-15

**Authors:** Lijun Wang, Qiang Zhou, Haiyang Yang

**Affiliations:** 1School of Materials Science and Engineering, Henan Joint International Research Laboratory of Nanocomposite Sensing Materials, Anyang Institute of Technology, Anyang 455000, China; 2CAS Key Laboratory of Soft Matter Chemistry, School of Chemistry and Materials Science, University of Science and Technology of China, Hefei 230026, China; zq199532@mail.ustc.edu.cn (Q.Z.); yhy@ustc.edu.cn (H.Y.)

**Keywords:** ring-opening alternating copolymerization, PEGylated polyester copolymer, rhodamine, fluorescent pH nanosensor, lysosomal pH detection

## Abstract

A novel lysosome-targeting PEGylated polyester-based fluorescent pH nanosensor is fabricated by the combination of ring-opening copolymerization (ROCOP), side-group modification and subsequent self-assembly. First, a key target amphiphilic copolymer carrier for rhodamine (Rh) pH indicator is synthesized in a facile manner by the ROCOP of phthalic anhydride with allyl glycidyl ether using mPEG-OH and *t*-BuP_1_/Et_3_B as the macroinitiator and binary catalyst, respectively. Subsequently, Rh moieties are covalently attached on the polymer chain with controllable grafting degree via an efficient thiol-ene click reaction. Concurrently, the effect of catalyst systems and reaction conditions on the catalytic copolymerization performance is presented, and the quantitative introduction of Rh is described in detail. Owing to its amphiphilic characteristics, the rhodamine-functionalized polyester-based block copolymer can self-assemble into micelles. With the covalent incorporation of Rh moieties, the as-formed micelles exhibit excellent absorption and fluorescence-responsive sensitivity and selectivity towards H^+^ in the presence of various metal cations. Moreover, the as-prepared micelles with favorable water dispersibility, good pH sensitivity and excellent biocompatibility also display appreciable cell-membrane permeability, staining ability and pH detection capability for lysosomes in living cells. This work provides a new strategy for the facile synthesis of novel biocompatible polymeric fluorescent pH nanosensors for the fluorescence imaging of lysosomal pH changes.

## 1. Introduction

Lysosome, as one of the important intracellular digestive organelles, serves as a major degradation compartment for the molecular degradation behaviors during cell metabolism [1,2,3]. In particular, lysosomal pH plays the critical role in cellular metabolic processes, including apoptosis [4], autophagy [5] and cell maturation [4]. It has become an important indicator for physiological research, and has a significant effect on the cellular functions and disease-prevention [6]. When lysosomes maintain acidic pH in the range from 4.0 to 5.5, the interior hydrolytic enzymes are activated, which degrade macromolecules and cell components, whereas abnormal lysosomal pH always leads to pathological cellular dysfunction, inducing an increased risk of several diseases [7]. Therefore, the quantitative detection of lysosomal pH changes is of vital importance for cellular analysis or diagnosis. Recently, fluorescence methods showing superior advantages of noninvasiveness, highly spatial resolution, high sensitivity, simple operation, real time and in situ detection, have been widely used in tracking lysosomal pH changes [8,9,10,11,12]. To date, a number of fluorescent sensors have been well developed for this purpose [13,14,15,16]. Despite extensive efforts, most of the reported sensors are based on small organic dye derivatives, such as rhodamine [13,14,17], fluorescein [18,19,20], cyanine [21,22], pyrene [23,24] and naphthalimide [15,25], etc. However, the classical fluorescent small-molecule sensors always suffer from less satisfactory water solubility and poor photostability [26]. Moreover, most exhibit one single emission band, and the corresponding detection result can usually be affected by dye concentration, photobleaching property and fluctuation of instrumental parameters, etc. [16,25,27] Alternatively, polymeric fluorescent nanosensors formed by functional amphiphilic copolymers have thus been explored recently as promising candidates to circumvent these restrictions. In comparison to small organic dye sensors, polymeric fluorescent nanosensors exhibit several advantages, including tunable structure design, good water dispersibility, large accumulation capacity, long in vivo circulation times, facile handling, high photostability, excellent biocompatibility and signal amplification [16,28,29,30,31]. Thus far, various polymeric fluorescent pH nanosensors with high sensitivity and selectivity for tracking the pH changes in lysosomes have been well developed, and most reported systems are fabricated by covalently incorporating weakly acidic pH-sensitive fluorescent organic dyes into amphiphilic polymer [6,7,29,32]. The organic dyes can endow the nanosensors with fluorescence intensity-changeable ability with the response to the variation of pH. Meanwhile, the amphiphilic polymer can self-organize into nanoparticles in aqueous solution, thus guaranteeing the incorporated organic dyes with enhanced water solubility and photostability [7,16,29,33].

Although strenuous efforts have been focused on the development of polymeric fluorescent lysosomal pH nanosensors, and most of the nanosensors are efficient with high resolution for lysosomal imaging [6,7,15,17,34,35,36,37,38], the commonly reported preparation strategies are still not satisfactory for the instable grafting degree of fluorophore, the complexity of the preparation process or the potential cytotoxicity of the polymer carriers. Typically, polymeric fluorescent nanosensors are always fabricated by the free-radical copolymerization of sensitive fluorescent vinyl monomers and parent comonomers [6,7,15,17,34,35], or the ring-opening polymerization of cyclic ester monomers and subsequent post-modification with pH-sensitive fluorescent moieties [37,38]. The former synthetic strategy apparently can hardly control the stability of the grafting degree of fluorophore, and requires the strict control of polymerization conditions; at the same time, the resultant polymer carriers always exhibit low biocompatibility [39]. On the other hand, the polymeric nanosensors synthesized by the latter approach present several attractive features, such as stability of the grafting degree, biocompatibility, cell permeability and biodegradability. However, some limitations with regard to the latter synthetic strategy still need to be considered. First, there are few commercially available cyclic ester monomers; most of the cyclic ester monomers are not available and need to be prepared in advance. Second, most of the cyclic lactone monomers do not have any functionality; thus, these monomers can barely tolerate available fluorescent functionalized modification of parent polymers, which is essential to prepare polymeric fluorescent nanosensors [39,40,41]. Therefore, it is still challenging to develop a more appropriate alternative approach for the facile synthesis of biocompatible lysosome-targeted pH-sensitive fluorescent polymeric nanosensors with good stability of grafting degree for academic research and practical applications.

Very recently, the epoxide-anhydride ring-opening copolymerization (ROCOP) catalyzed by metal-free Lewis pairs has attracted considerable attention. Due to the advantageous features of this ROCOP, such as atom-economic, enriched monomer resources and mild reaction conditions, it has become an effective and promising strategy for the synthesis of biocompatible polyesters [42,43,44,45]. By taking advantage of this versatile ROCOP, various polyesters with tunable structures, functionalities and good biocompatibility can be successfully synthesized [46,47,48,49]. In fact, based on this type of ROCOP, a serials of functionalized polyester-based amphiphilic polymers have been well developed. With properties of facile synthesis, good biocompatibility and biodegradability, the polyester-based amphiphilic polymers have been successfully used in a wide variety of biomedical fields [50,51,52]. Despite these encouraging advantages, the metal-free Lewis pairs catalyzed ROCOP of epoxides and anhydrides have never been used for the synthesis of polyester-based fluorescent pH nanosensors specially for lysosome pH detection.

In other words, although the polymeric fluorescent pH nanosensors always present many advantageous properties compared with small molecular sensors, most reported systems are still not satisfactory for their imperfect preparation strategies, low biocompatibility and poor biodegradability. In comparison with the most reported methods to prepare polymeric fluorescent pH nanosensors through the abovementioned radical copolymerization or ROAC of cyclic lactone monomers, the ROCOP of epoxides and anhydrides may presents many advantages, such as facile preparation, wide availability of raw materials, flexible design, polymer structure with tunable functionalities, etc. [41,44,47] Meanwhile, the fabricated polymeric pH nanosensors via the ROCOP of epoxides and anhydrides also exhibits many benefits over others, including good biocompatibility, good cell permeability, better potential biodegradability and tunable structure and function, etc. [41,44,47] Thus, if the ROCOP of epoxides and anhydrides, hydrophilic modifications and decoration with fluorescent dyes can be realized together, the attractive polymeric pH nanosensors with advantageous properties may be fabricated conveniently and effectively by a novel approach. In fact, the fabrication of the polyester-based biocompatible polymeric fluorescent pH nanosensors might significantly promote the development of novel lysosome fluorescent probes. Additionally, selecting proper acidic pH-sensitive fluorescent organic dyes also plays a key role in the successful fabrication of highly effective pH nanosensors. Although a large amount of specific lysosomal probes have been reported [13,17,34], these probes always can induce an increase of pH value in the acidic compartments, leading to high nonspecific background fluorescence signals inside cells. By contrast, the rhodamine dyes extensively employed as fluorescent pH probes for acidic environments can effectively prevent the interference of background fluorescence favored by the H^+^-induced ring-opening mechanism, and have been commonly reported as an ideal choice to imaging lysosomal [17].

On the basis of the above introduction, we propose a concept for the design and synthesis of lysosome-targeting polyester-based amphiphilic polymeric fluorescent pH nanosensors by the ROCOP of epoxide with anhydride catalyzed by metal-free Lewis pairs, a thiol-ene click reaction and subsequent self-assembly. Rhodamine was selected as the fluorescence organic dye due to its highly fluorescent pH-sensitivity [53,54,55,56]. Concurrently, biocompatible polyester-based amphiphilic polymers with abundant alkenyl functional groups, serving as the polymer carriers, can covalently attach rhodamine moieties with controllable grafting degree via an efficient thiol-ene reaction. Notably, the catalytic copolymerization performance of epoxide with anhydride was evaluated under various conditions. Subsequently, the successful covalent incorporation of rhodamine fluorophores together with the substitution degree were estimated. Furthermore, the self-assembly behaviour of the rhodamine-functionalized polyester-based amphiphilic copolymer, selectivity and sensitivity of polymer micelles toward H+ in aqueous solutions, and its potential application in fluorescent pH nanosensors for tracking pH changes in lysosomes were systemically evaluated (Figure 1).

## 2. Materials and Methods

### 2.1. Materials

All manipulations were carried out in a nitrogen-filled glove box. mPEG-OH (5000 g/mol, Aladdin, https://www.aladdin-e.com/) was dried in vacuum at 60 °C for 24 h prior to use. Allyl glycidyl ether (AGE, 98%, Aladdin, https://www.aladdin-e.com/) was distilled after being dried over CaH_2_. Phthalic anhydride (PA, 99%, Aladdin, https://www.aladdin-e.com/) was purified by subliming several times under reduced pressure. Triethyl borane (Et_3_B in THF, 1.0 mol/L, Aladdin, https://www.aladdin-e.com/), t-BuP_1_ (98%, Aldrich, https://www.sigmaaldrich.cn/CN/zh), 2,2′-Azobis(2-methylpropionitrile) (AIBN, Aladdin, https://www.aladdin-e.com/) were all used directly without purification. N-(2-(3′,6′-bis(diethylamino)-3-oxo-4a′,9a′-dihydrospiro[isoindoline-1,9′-xanth-en]-2-yl)ethyl)-2 mercap-toacetamide (RhB-SH) was synthesized according to the literature procedures [57], Bafilomycin A1 (ACROS, https://www.acros.com/).

### 2.2. Characterization

NMR spectra were recorded at 25 °C on a Bruker AV400 NMR spectrometer (Bruker, https://www.bruker.com/zh.html) using deuterated chloroform (CDCl_3_) as the solvent and tetramethylsilane as the internal standard. Molecular weights and molecular weight distributions of polymers were determined with a Viscotek GPCmax system (Malvernpanalytical, https://www.malvernpanalytical.com.cn/) consisting of a pump, a Viscotek UV detector, and a Viscotek differential refractive index (RI) detector. The size exclusion chromatography (SEC) was conducted in THF at 1.0 mL/min at 35 °C. The absorption spectra were collected on a UV-2600 spectrophotometer (Shimadzu, https://www.shimadzu.com.cn/). Fluorescence spectra were recorded using a F-7000 spectrofluorometer (Hitachi, https://www.hitachi.com.cn/about/network/group/hyq-sh.html). The slit widths were set at 5 nm for both excitation and emission, and the λ_ex_ = 365 nm. The hydrodynamic diameter and size distribution of the polymeric micelles were measured by dynamic light scattering (DLS) at a scattering angle θ of 90° as a function of temperatures by using a 90 Plus Particle Size Analyzer (Malvernpanalytical, https://www.malvernpanalytical.com.cn/). The TEM morphologies of the produced polymeric micelles were observed by transmission electron microscopy (TEM) on a JEM 1230 electron microscope (Jeol, https://www.jeol.co.jp/) operated at an acceleration voltage of 100 kV.

### 2.3. Preparation

#### 2.3.1. Representative Copolymerization Procedure

Typically, mPEG-b-P(PA-alt-AGE) was synthesized by ROCOP of AGE with PA using mpeg-OH (5000 g/mol) as the initiator. In a glovebox, AGE (1.186 mL, 0.01 mol), PA (1.48 g, 0.01 mol) and mpeg-OH (0.50 g, 0.10 mmol) were dissolved in 4.0 mL THF firstly, then added into a flask. Thereafter, Et_3_B in tetrahydrofuran solution (40 μL, 0.04 mmol) and *t*-BuP1 (10 μL, 0.04 mmol) were added by vigorous stirring. Finally, the flask was taken out from the glovebox in a sealed state, and subsequently placed in a 90 °C oil bath. After a desired reaction time, the copolymerization was stopped before adding 1.0 mL of acetic acid to quench it, and then the mixture solution was precipitated into cold ether multiple times. The obtained copolymer was then collected and dried in vacuum.

#### 2.3.2. Thiol-Ene Click Reaction between mPEG-b-P(PA-alt-AGE) and RhB-SH

Typically, the synthesis of rhodamine-functionalized amphiphilic copolymer was started with the ratio of reagents [C=C]_0_/[RhB-SH]_0_/[AIBN]_0_ = 1/0.3/0.1. The click reaction between the PEGylated polyester block copolymer (0.1 mmol of C=C group) and RhB-SH (0.03 mmol) were conducted under a nitrogen atmosphere. The reaction materials were first dissolved in 4 mL THF with AIBN as initiator. Then, the mixture was placed in a 70 °C oil bath and was continuously stirred for 4h. Afterwards, the reaction was stopped, the solution was filtered and the solvent was removed under vacuum. THF was used to dissolve the crude products, which were then precipitated into cold diethyl ether twice, and the final reaction products were collected and vacuum-dried at 45 °C overnight.

#### 2.3.3. Preparation of Rhodamine-Functionalized Amphiphilic Copolymer Micelles

A total of 50 mg of rhodamine-functionalized copolymer was first dissolved into 3 mL THF, then dropped into 25 mL deionized water, and stirred for about 1 h. Subsequently, THF was removed through dialysis against deionic water. After dilution, the aqueous solution in the dialysis bag was subjected to DLS measurement. TEM measurements were also carried out on the transmission electron microscope.

#### 2.3.4. Cell Imaging

To evaluate the imaging of lysosomal pH changes, HeLa cells were firstly pretreated with various concentrations (0, 40, 80, 100 nM) of bafilomycin A1 for 9 h, then incubated with an mPEG-b-P(PA-alt-AGERh) nanosensor (5 μM) for 6 h at 37 °C. After being cultured for 24 h, each plate received 500 μL fresh medium enriched with bafilomycin A1. After incubation for 9 h, the plates were purged with fresh PBS, then incubated for additional 6 h with 5 μM nanosensor. Afterwards, fluorescence imaging was performed after HeLa cells were again washed with fresh PBS buffers (pH = 7.0) and incubated with fresh medium. On a fluorescence microscope (Leica, Germany, https://leica-camera.com/en-int), all images were taken with an an oil immersion objective. Excitation: 560 nm, red emission collected: 560–700 nm.

To evaluate the imaging of co-localization, Hela cells were cultured with fresh medium containing polymeric nanosensors (50 μg/mL) for 3 h, then stained with Lyso Tracker Green DND-189 (1 μM) and incubated for 30 min. Thereafter, the cells were washed in PBS and imaged using a Zeiss 7103-channel confocal laser scanning microscopy (CLSM) (Zeiss, Germany, https://www.zeiss.com.cn/gongyeceliang/guanyuwomen/shengchanjidi/bochingen.html). Channel 1 (Lyso Tracker): excitation: 488 nm, emission collected: 488–560 nm; Channel 2 (polymeric nanosensors): excitation: 560 nm, emission collected: 560–700 nm.

#### 2.3.5. Cytotoxicity of Rhodamine-Functionalized Polymer Micelles

The cytotoxicity of the rhodamine-functionalized polymer micelles was determined with Hela cells. Optical microscopy (Leica) was used to examine the morphology of the cells. MTT analysis was used to evaluate the cell viability of the micelles. In detail, cells were seeded in 160 μL of appropriate medium containing 10% FBS in 96-well microplates. The cells were incubated with different concentrations of micellar nanoparticles (0, 10, 30, 50, 70 and 100 μg mL^−1^). Then, the micelles were removed by washing the cells with fresh PBS repeatedly. Afterwards, 20.0 μL of MTT stock solution was added into each well and cultured for another 4h. After that, the plates were examined using a microplate reader, and 150.0 μL of DMSO was replenished. Subsequently, the system was shaken for several minutes, and the absorbance of the system was measured at 390 nm using a Bio-Rad reader (Bio-Rad, https://www.bio-rad.com/). The experiments were repeated three times.

## 3. Results and Discussion

### 3.1. Synthesis of mPEG-b-P(PA-alt-AGE)

To synthesize the target amphiphilic polymers, we first aim to prepare an amphiphilic copolymer, mPEG-b-P(PA-alt-AGE), with abundant alkenyl groups endowing the copolymer with further functionality by post-modification. The mPEG-b-P(PA-alt-AGE) was synthesized by ROCOP of PA and AGE using mPEG-OH as the initiator and t-BuP1/Et3B as the binary catalyst (Figure 2). Since this catalytic copolymerization behaviour has rarely been reported, the related copolymerization performance is worth investigating. Thus, the impact of reaction conditions on the copolymerization performances of PA and AGE initiated by mPEG-OH was initially investigated and evaluated. Specifically, the copolymerization results corresponding to different feed ratios, reaction time and reaction temperatures were collected and analyzed (Table 1). In addition, the polymerization condition was expected to be optimized to realize effective copolymerization with perfect chemoselectivity and high reactivity.

The copolymerization of PA with AGE was conducted in THF, with a targeted degree of polymerization being 40 (DP = 20 per hydroxyl). After the reaction was conducted for a certain time, the viscosity of the mixture increased noticeably, suggesting the apparent molar mass of the product progressively increases, indicating the occurrence of polymerization. The copolymerization data are summarized in Table 1, and the SEC traces for the synthesized polymers obtained from the isolated products are shown in Figure 1. As seen in Table 1, the molecular weights (both M*_n,NMR_* and M*_n,SEC_*) of the corresponding PEGylated polyester indeed increased with prolonged polymerization time (Table 1, entries 1, 2 and 5), indicating the increase of monomer conversion. Notably, as the conversion of monomer increased, the molar mass distribution of polyester product become narrow. When the reaction time reached 48 h, the product maintained an advantageous unimodal molar mass distribution. Moreover, various copolymerization temperatures were investigated from 45 °C to 90 °C (Table 1, entries 3–5). The results indicate the copolymerization proceeded successfully at all the investigated temperatures.

It demonstrated that the copolymer synthesized at 90 °C displayed a higher PA conversion and the highest Mn. Additionally, feeding various equiv. of *t*-BuP_1_ versus BEt_3_ led to different copolymerization results (Table 1, entries 5–7). Although the resultant copolymers displayed similar number-average molecular weights (M*_n,NMR_*), their SEC traces were quite different from each other; only when the feed ratio of *t*-BuP_1_ to BEt_3_ was 0.2:0.4, the corresponding polymer maintained an unimodal molar mass distribution (Table 1, entries 5–7; Figure 1). As discussed above, the t-BuP1/Et3B can effectively catalyze the ROCOP of PA and AGE by using mPEG-OH as the macroinitiator, and when the feed ratio of PA/AGE/mPEG-OH/t-BuP1/Et3B was set as 20:20:1:0.2:0.4, the temperature was 90 °C and the reaction time reached 48 h, and an optimized mPEG-b-P(PA-alt-AGE) without metal residues with a higher M*_n,SEC_* of 10.6 kDa and proper Đ values of 1.28 would be obtained (Table 1, entry 5).

The structure of the mPEG-b-P(PA-alt-AGE) was confirmed by NMR characterization. The ^1^H NMR and ^13^CNMR spectra (Table 1, entry 5) are shown in Figure 2A and Appendix A, where all the characteristic proton and carbon signals are well-assigned in accordance with the predicted copolymer structure. The signal peaks appearing at 5.5 and 4.6 ppm in the ^1^H NMR corresponded to the protons of the methylene groups next to the newly formed ester linkages after copolymerization. Besides, the signal peaks at about 5.85 and 5.2 ppm were attributed to the alkenyl group, endowing the synthesized copolymer with the ability to be further functionalized by click reaction. Notably, taking advantage of this ROCOP, if the monomer was replaced by other functional ones, novel functional groups or post-modifying groups could be introduced; thus, this copolymerization technique can guarantee the copolymer with tunable functionalities. Compared with the similar previous works reported by Wang [34] or Yu [17], the structure of the copolymer fabricated here presents better controllability and designability.

Moreover, by comparing the integral of proton peak at 5.58 from AGE monomer with the proton peaks at 7.5–7.8 ppm from PA monomer, the molar ratio of PA to AGE on the copolymer backbone was calculated to be 1:1, confirming the formation of an alternating structure of the P(PA-alt-AGE) segment. Furthermore, by comparing the integral of proton peak “a” from the residual groups of the initiator with the proton peak “g” from PA monomer (Figure 2A), the polymerization degree of the P(PA-alt-AGE) segment was calculated to be 33 (for PA or AGE monomer, DP = 16.5 per hydroxyl). Since the targeted degree of polymerization was 40, the actual monomer conversion was more than 80%. Correspondingly, the M_n,NMR_ of the copolymer was calculated to be 9200. In general, the above results suggest the successful synthesis of mPEG-b-P(PA-alt-AGE) block copolymer with a proper Mn and pendant alkenyl functional groups.

### 3.2. Postpolymerization Modification of mPEG-b-P(PA-alt-AGE) and Self-Assembly

To prepare rhodamine-functionalized polyester-based amphiphilic polymer (mPEG-b-P(PA-alt-AGERh)), a standard thiol-ene click reaction was carried out to conjugate side alkenyls with RhB-SH and to control the grafting degree of rhodamine. The RhB-SH (Figure 2B), which has ever been reported as pH- or metal cations fluorescent indicator [13,58],was chosen as the potential pH-responsive fluorescent reagents for the further modification of mPEG-b-P(PA-alt-AGE) (Table 1, entry 5). The chemical structure of this target mPEG-b-P(PA-alt-AGERh) was identified via ^1^H NMR and ^13^C NMR spectroscopy (Figure 2C,D). The attenuation of the alkenyl proton resonance peaks at 5.85 or 5.2 ppm and the appearance of proton peaks from the rhodamine moieties (peaks a, d, e, l and o) indicated the successful conjugation of rhodamine onto the side chains of mPEG-b-P(PA-alt-AGE) (Figure 2C). Additionally, the ^13^C NMR spectrum confirmed the success of the click reaction (Figure 2D). Moreover, the SEC analysis indicates the change of the polymer molar mass from 10.6 kDa to 11.2 kDa after the conjugation of rhodamine, further verifying the success of the click reaction (Figure 3A). Importantly, by comparing the integral of proton peak “j” or “f” with the proton peak “d” (Figure 2C), the grafting degree of rhodamine attached on the P(PA-alt-AGE) was calculated to be about 0.3, meaning about 0.3 eqv. of side alkenyls on the P(PA-alt-AGE) were functionalized with rhodamine, which was in accordance with the pre-designed grafting degree proposed in the experimental part, and verified the controllability of the grafting degree of rhodamine.

Alternatively, the percentage of rhodamine attached to the polyester backbone was also estimated by using a standard curve method. Specifically, the UV-vis absorption spectra of RhB-SH in DMF solution with different concentrations were firstly examined; then, the ratio of the characteristic absorption of the ring-closed RhB-SH at 316 nm was plotted versus the RhB-SH concentrations (Figure 3B) and, thus, the linear relationship between the characteristic absorption peak intensity and RhB-SH concentrations was obtained (Figure 3C). Afterwards, the absorption spectra of RhB-SH, mPEG-b-P(PA-alt-AGERh) and mPEG-b-P(PA-alt-AGE) in DMF solution were measured (Figure 3D). From Figure 3D, it can be seen that both mPEG-b-P(PA-alt-AGERh) and RhB-SH showed absorbance at 316 nm, which were attributed to the characteristic absorption of rhodamine in a ring-closed form. In fact, the introduction of a polymer structure did not change the Uv-vis absorption characteristics of Rh moieties; thus, mPEG-b-P(PA-alt-AGERh) and RhB-SH showed a similar absorption characteristic peak at 316 nm [34]. Therefore, according to the linear relationship between the characteristic absorption peak intensity and RhB-SH concentrations, the mass concentration of the RhB-SH immobilized to the copolymer was calculated to be 20.24%, and the corresponding grafting degree of rhodamine conjugated onto mPEG-b-P(PA-alt-AGE) was about 0.27 (molar concentration), which was consistent with the target grafting degree described in the post-modification reaction recipe, and again confirmed the controllability of grafting rhodamine onto the mPEG-b-P(PA-alt-AGE) through thiol-ene click reaction.

Furthermore, the self-assembly behaviour of the synthesized mPEG-b-P(PA-alt-AGE) in aqueous solution was investigated. Dynamic laser light scattering (DLS) analysis revealed hydrodynamic diameters *D*_h_ of 119 nm (Appendix A). Concurrently, TEM images also showed an average size of about 90 nm (Appendix A). The above results effectively confirmed the formation of a micellar structure and verified the amphiphilicity of the copolymer. Furthermore, after post-modification, the as-formed Rh-functionalized copolymer still exhibited amphiphilic property; thus, it could also form micelles in aqueous solution. The copolymer micelles formed in aqueous solution are spherical in shape. The dark cores were associated with the aggregation of the hydrophobic rhodamine-functionalized polyester, and the shells surrounding the nanomaterials were attributed to the crystallization of the PEG block chains during the shrinking of the micelles with the evaporation of the solvent. The critical micelle concentration (CMC) of the self-assembled micelles was determined to be 0.08348 mg mL^−1^ by fluorescence measurements using pyrene as the probe (Figure 4A). The micelle size was investigated by DLS, and the result revealed the micelles exhibited an average diameter of roughly 90 nm with a narrow size distribution (Figure 4B). The mPEG-b-P(PA-alt-AGERh) micellar nanoparticles maintained stability for more than 4 weeks (Appendix A), indicating the micelles formed in aqueous media were pretty stable. The transmission electron microscopy (TEM) images in Figure 4C showed that the micelles exhibited spherical morphology and the diameter was about 70 nm. The abovementioned results indicate that the fabricated mPEG-b-P(PA-alt-AGE-Rh) micelles exhibited a low cmc, a proper average diameter less than 100 nm and a narrow size distribution, which might be beneficial for the application of the mPEG-b-P(PA-alt-AGE-Rh) micelles in tracking the pH changes in living cells.

### 3.3. pH-Sensing Properties of mPEG-b-P(PA-alt-AGERh) Micelles

Rhodamine, as one kind of classical fluorescent dyes, exhibits outstanding photophysical properties, and has been widely used as a fluorescent pH nanosensor for acidic environments. The spirolactam structure of rhodamine derivatives always exhibits sensitivity to pH variations, and can undergo a pH-dependent equilibrium between the non-fluorescent (colorless) ring-closed form and the fluorescent (colored) ring-opened form. When the situation is basic or neutral, the rhodamine remains closed, and the system is non-fluorescent; whereas the acidic situations always induce the ring-opening of spirolactam and the system exhibits strong fluorescence and a pink-to-red color (Figure 1) [17,19,34]. Furthermore, when the rhodamine derivatives were incorporated into the hydrophobic core of the polymeric micelles, they also presented excellent pH sensitivity [17,34].

The responsiveness of the mPEG-b-P(PA-alt-AGERh) micelles as nanosensors towards pH was investigated by UV–vis absorption and fluorescence spectroscopies. Figure 5A,B presents the absorption spectra and fluorescence spectra of mPEG-b-P(PA-alt-AGERh) micelle solutions at different pH values. The maximum absorption peak and emission peak (λ_ex_ = 365 nm) are at 560 and 582 nm, respectively, which were attributed to the formation of the ring-opened rhodamine. As shown in Figure 5A,C, the absorption spectra showed nearly no changes when the pH value > 6.0, which was ascribed to the non-obvious ring-opening of rhodamine, whereas when decreasing pH to <6.0, the absorption intensity of polymer micelles at 560 nm increases, indicating the formation of the ring-opened spirolactam form of rhodamine moieties in polymeric nanosensors. Concomitant with the emergence of the characteristic absorption band, the apparent color change can be observed from colorless to pink (Figure 6A). Furthermore, the fluorimetric response of the polymeric micelles with different concentrations of H^+^ was also investigated (Figure 5B,D). The pH-dependent fluorescence emission spectra of mPEG-b-P(PA-alt-AGERh) micelles agreed well with the abovementioned pH-dependent absorption spectra. The micelle solution emitted no fluorescence at 550–600 nm when the pH was above 6.0, whereas a considerably enhanced fluorescence signal centered at 582 nm appeared with the decrease of pH from 6.0 to 2.0. Correspondingly, the fluorescence images of the micelle solutions under UV light (365 nm) showed fluorescent color changes from blue to pink with decreasing pH from 12.0 to 2.0 (Figure 6B). The abovementioned fluorimetric responsive characteristics of the polymeric micelles further demonstrates the formation of ring-opened rhodamine induced by H^+^ under acidic conditions. The absorption and emission changes associated with the H^+^-induced spirolactam ring opening of the rhodamine group demonstrates the responsiveness of the mPEG-b-P(PA-alt-AGERh) micelles as fluorescent nanosensors towards pH, and further indicates their suitability for studying acidic organelles.

Furthermore, to evaluate the fluorescent sensitivity of the mPEG-b-P(PA-alt-AGERh) micelle nanosensors to acidic pH, we also investigated the dynamic processes of fluorescence responsive behaviours of the polymer micelles induced by H^+^. The response time of the polymer micelles towards H^+^ was recorded systematically at a pH of 3.0 (Figure 7A). As shown in Figure 7A, the fluorescence intensity of polymer micelles at 582 nm increased rapidly after H^+^ was added to the micelle solutions in the first 5 min, then further increased in the subsequent 55 min afterwards, gradually increased over the course of 60 min and stabilized thereafter. Meanwhile, to gain further insight into the fluorescence intensity enhancement kinetics, the fluorescence intensity ratio was plotted against time and fitted to a single-exponential decay equation *N*(*t*) = *N*_0_ + *N*_1_e^–t/*τ*^ with different time constants (Figure 7B). The time constant, *τ*, calculated from the fluorescence change, was about 36 min. The rapid response time was short as 5 min; the small response time constant and a longtime responsive stability indicated that the mPEG-b-P(PA-alt-AGERh) micelles are indeed sensitive and are effective nanosensors for acidic pH.

The selectivity of the mPEG-b-P(PA-alt-AGERh) micelle solutions for pH was further investigated by fluorescence measurements. The representative fluorescence spectra of the copolymeric micelles (1.5 mg·mL^−1^) in tris-buffer solutions (pH 7.0) after the addition of various cations (100 μM) are shown in Figure 7C. It can be seen from Figure 7C,D that no visible fluorescence enhancement can be observed with the addition of various cations, including K^+^, Zn^2+^, Ba^2+^, Al^3+^, Mn^2+^, Ni^2+^, Co^2+^, Cd^2+^, Cu^2+^, Hg^2+^, Fe^3+^, Mg^2+^, Ca^2+^, Pb^2+^, Sr^2+^, Cr^2+^ and Gd^2+^, in comparison with a blank sample (tris-buffer solutions, pH 7.0). Whereas only when the micelle solutions were acidic, the solution emitted obviously strong fluorescence centered at 582 nm. Accordingly, the impact of cations on the absorption characteristics of the mPEG-b-P(PA-alt-AGERh) micelles agreed well with the fluorescence characteristics. As shown in Appendix A, the micelle solutions of mPEG-b-P(PA-alt-AGERh) exhibited no visible absorbance enhancement after the addition of metal ions when the pH value was 7.0. Notably, the absorbance in presence of Fe^3+^ was slightly stronger; however, it was only associated with the formation of small iron (III) clusters [59], but had nothing to do with the ring-opening of rhodamine. The abovementioned conjecture was in accordance with the fluorescence results shown in Figure 7C, and no visible fluorescence enhancement can be observed with the addition of Fe^3+^. Whereas the pH value of the micelle solutions was below 6.0, the solutions showed considerable absorption characteristics, which was in accordance with the visible color change (Appendix A). The above results verified that H^+^ could selectively lead to spirolactam ring opening of rhodamine, and induce the dramatic emission or absorbance enhancement. Particularly, the fluorescence response processes of the mPEG-b-P(PA-alt-AGERh) micelles exclusively towards H^+^ in the presence of various cations proved that it can serve as a highly selective fluorescence nanosensor for H^+^ and is favorable to work as a pH indicator.

### 3.4. Fluorescence Imaging and Lysosome Staining in Living Cells

To examine the potential application of the nanosensor in biological systems, we investigated its fluorescence imaging behaviour in living cells. Initially, HeLa cells were selected and incubated with various concentrations (0, 1.0, 5.0, 10.0 μM) of nanosensor for 6 h at 37 °C, respectively; red fluorescence of the nanosensor within the cells was observed when the concentration of nanosensor was relatively high (Figure 8), demonstrating that the polymeric nanosensor was capable of staining living cells.

Furthermore, to confirm the distribution of the polymeric nanosensor in the cell, a classic lysosome tracker, LysoSensor^®^ Green DND-189, was selected to co-stain HeLa cells with the nanosensor (Figure 9). HeLa cells were first incubated with 5 μM nanosensor for 3 h, then further incubated with 1 μM LysoSensor^®^ Green for 1 h. As seen in images collected with confocal microscopy (Figure 9), it can be found that the distribution of nanosensor with red emission colocalized with LysoSensor Green with green emission, implying that the nanosensor distributed mainly in the lysosome. Therefore, the nanosensor could selectively stain lysosome in living cells. Moreover, the cytotoxicity of the nanosensor on HeLa cells was also studied (Appendix A). HeLa cells were incubated with polymeric nanosensor for 8 or 24 h, and the cell viability was measured by MTT assays. As shown in Appendix A, the nanosensor proved basically non-toxic to HeLa cells under the experimental conditions, and the bio-friendly PEG and polyester chains all contributed to improving the biocompatibility of the nanosensor. In comparison with a similar polymeric sensor fabricated previously through free-radical copolymerization [17,34], the PEGylated polyester-based copolymeric sensors exhibited better biocompatibility.

### 3.5. Detection of Lysosomal pH Changes in Living Cells

Bafilomycin A1, a classical inhibitor of the vacuolar-type H^+^-ATPase (V-ATPase), could effectively inhibit lysosomal acidification and then further induce the increase of the lysosomal pH. Taking advantage of the ability of bafilomycin A1 to adjust the lysosomal pH, we then proceeded to investigate the possibility of the mPEG-b-P(PA-alt-AGERh) nanosensor for the detection of lysosomal pH. HeLa cells were first incubated with various concentrations (0, 40, 80, 100 nM) of bafilomycin A1 for 9 h at 37 °C, then with 5 μM mPEG-b-P(PA-alt-AGERh) nanosensor for 6 h. With the increase of the concentration of bafilomycin A1, the fluorescence intensities within the cells obviously weakened (Figure 10). From the red fluorescence decrement after being treated with bafilomycin A1, it can be demonstrated that the mPEG-b-P(PA-alt-AGERh) nanosensor possessed the potential capability to monitor the lysosomal pH changes. Thus, the polymeric nanosensor synthesized here can serve as a fluorescent indicator for imaging lysosomal pH changes in living cells.

## 4. Conclusions

In conclusion, we present a facile ring-opening copolymerization and side-group modification technique together with self-assembly for preparing novel lysosome-targeting PEGylated polyester-based fluorescent pH nanosensors. First, the target novel PEGylated polyester-based alternating copolymer carrier, viz. mPEG-b-P(PA-alt-AGE), with abundant alkenyl functional groups was facilely synthesized by the ROCOP of PA and AGE using mPEG-OH and *t*-BuP_1_/Et_3_B as the macroinitiator and binary catalyst, respectively. Concurrently, the impact of catalyst systems and polymerization conditions on copolymerization performance was systematically evaluated. Second, by taking advantage of the alkenyl functional pendants on the polyester segments, through the efficient thiol-ene reaction, the rhodamine moieties, as effective fluorescent pH indicators, were successfully covalently attached on the polymer chain with a controllable grafting degree. Owing to the amphiphilic characteristics, the obtained polyester-based amphiphilic copolymer could self-assemble into micelles in the aqueous solution with hydrophobic polyester backbone cores and hydrophilic PEG shells, and the micellar structure guarantees the Rh moieties with enhanced water dispersibility and photostability. Due to the covalently encapsulation of rhodamine dye, the formed polymer micelles showed excellent absorption and fluorescence responsive sensitivity towards pH. Furthermore, the micelles exhibited rapid responsiveness, and high selectivity toward H^+^ in the presence of various metal cations. These advantageous properties guarantee the polymer micelles the detection ability towards H^+^ as the polymeric fluorescent pH nanosensors. Moreover, the rhodamine-functionalized polyester-based amphiphilic polymer nanosensors also showed excellent biocompatibility with HeLa cells, and could selectively stain lysosome in the cells. With the advantageous properties of sensitivity and selectivity for pH, amphiphilic polymer nanosensors were also successfully used to visualize the intracellular lysosomal pH changes. In comparison with other sensors, the polyester-based amphiphilic polymer nanosensors fabricated here exhibit many benefits, including good biocompatibility, better potential biodegradability, tunable structure and function, etc. This work provides a new strategy to the facile synthesis of novel biocompatible polymeric fluorescent pH nanosensors for fluorescence imaging and detection of pH changes in living cells, and may inspire new facile approaches to prepare other biocompatible functional polymeric nanosensors.

## Data Availability

All data is contained within this article.

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
