# Peer review of "A Facile Fabrication of Lysosome-Targeting pH Fluorescent Nanosensor Based on PEGylated Polyester Block Copolymer"

_polymers, 2022, doi:10.3390/polym14122420_

Round 1
Reviewer 1 Report
The manuscript is in general very well written and presents a very interesting and well performed research on the development of a pH fluorescent nanosensor based on a PEG- and Rhodamine-functionalized polyester.
I only have a few concerns related mainly with the quality and size of some Figures, and some typing mistakes.
Scheme 1. Rhodamine at pH 7 appears blurred. This should be modified and improved.
Page 4, lines 180 and 181, Zeiss instead of Zesis?
Scheme 2. Should be amplified, in its current size is hardly readable on paper
Table 1. Improve the edition
Page 7, line 249, the peaks do not correspond strictly to the newly formed ester linkages, but to the protons of the methylene groups next to the ester linkages. This sentence should be revised and expressed correctly.
Figure 2. Should be amplified. It is not possible to analyze it on paper support.
Page 8, Line285 and Page 9, Line 324: "were" instead of "was"
Figure 4. Lines 332 and 333; What is MTG? Should it be AGE?
Figure 7C. I can only see 5 curves, not 15 ??
Figure 8. Line 426. Figure 10, Lines 463-464: "Row" instead of "column"?
Page 13, line 451. "pH detection in lysosomal"??
Reviewer 2 Report
The submitted manuscript describes a new lysosome-targeting PEGylated polyester-based fluorescent pH nanosensor fabricated by the combination of ring-opening copolymerization (ROCOP), side-group modification, and subsequent self-assembly. First, a key copolymer carrier for the rhodamine (Rh) pH indicator is synthesized in a simple route by the ROCOP of phthalic anhydride with allyl glycidyl ether using mPEG-OH and t-BuP1/Et3B as the macro-initiator and binary catalyst, respectively. The Rh moieties are covalently attached to the polymer chain with controllable grafting degree via an efficient thiolene click reaction. Concurrently, the effect of catalyst systems and reaction conditions on the catalytic copolymerization performance is presented, and the quantitative introduction of Rh is exhibited. With the covalent incorporation of Rh moieties, the as-formed micelles exhibit excellent absorption and fluorescence-responsive sensitivity and selectivity towards H+ in the presence of various metal cations. Moreover, the as-prepared micelles with favorable water dispersibility, good pH sensitivity, and excellent bio-compatibility also display appreciable cell-membrane permeability, staining ability and pH detection capability for lysosomes in living cells. Thus, I recommend this manuscript for publication after major revision.
Comments:
Introduction:
- Insert a new paragraph to explain the advantages of the used technique concerning other utilizing techniques.
- Clarify the benefits of pH nanosensor fabricated by the combination of ring-opening copolymerization (ROCOP) and make it the desired choice over others
- provide short notes about the difference between the studied mechanism with respect to other colorimetric or optical sensing mechanisms
Materials and methodology
- I suggest addressing the website of each supporting company (the links should be up to date) for both materials and instruments
- The data of the utilized instruments should be exhibited in details
Results and discussion
- The advantages of pH nanosensor fabricated by the combination of ring-opening copolymerization (ROCOP) in comparison with other methods should be highlighted, including analytical characteristics, reproducibility, specificity, stability
- Provide more details about the utilized mechanism?
- I suggest extending the pH measurements by the following values
2-2.5-3.0-3.5-4-4.5-5.0-5.5
- 4- The interfering substances are limited can you add more interfering metal ions? Especially Pb(II), Mn(II), Cr(II), Gd(II), Sr(II)…..etc.
- From Figure S4, the absorbance of Rhodamine labeled polymer at pH 5 and in the presence of Fe (III) are the same?
As we can observe the absorbance in presence of Fe(III) is scattered beam due to the agglomeration this is not really absorbance peak?? Needs comment with comparing these results with the fluorescence measurements
- TEM images of rhodamine-functionalized mPEG-b-P(PA-alt-AGERh) copolymer micelles ( the dark points and the shell surrounding the nanomaterials should be explained?)
- Fig 6 put the pH values inside the photo!
Reviewer 3 Report
The authors synthesised lysosome-targeting polyester-based amphiphilic polymeric fluorescent pH nanosensors by the ROCOP of epoxide with anhydride catalyzed by metal-free Lewis pairs. The results are interesting and the graphical abstract well presented the study. Some comments must be addressed before acceptance:
-Figure S1 has a low resolution and is hard to read
-In Fig S2 B, the particles are not clear and I don’t see any stable particle, should be replaced with a more resolution image or higher magnification
-the title can be shortened, please revise it
-some references are out of date, recently some interesting papers were published in biosensor science. Authors can use the following: https://doi.org/10.1002/aelm.202100233, https://doi.org/10.3390/bios12050314, https://doi.org/10.1016/j.jsamd.2022.100430, https://doi.org/10.1016/j.ijbiomac.2022.02.082, https://doi.org/10.1016/j.ijbiomac.2022.02.082
-the conclusion is too long; it needs to be summarized to present the final conclusion of the study
-why did the authors choose mPEG-b-P(PA-alt-AGE) as a copolymer, and what is the main advantage of this copolymer?
-it needs to compare your results with some similar studies and discusses in a more detailed manner
Round 2
Reviewer 2 Report
1- Authors reply to all the required comments significantly.
2- Now, this research is well arranged, with a sequence of clear ideas and concise writing that fits the research plan and methodology. The literature review is good, and they were able to successfully discuss their progress from both a perspective and an applied perspective. Their chosen method makes this data analysis excellent research and enables them to answer research questions and test their hypotheses. Thus, I strongly recommend this manuscript for publication in Polymers.
Author Response
Thanks for your comments.
Reviewer 3 Report
The paper can be accepted
Author Response
Thanks for your comments.